# Long-Term Results of Anodic and Thermal Oxidation Surface Modification on Titanium and Tantalum Implants

**Gabor Tamas Pinter** [1,*], **Balint Trimmel** [2], **Marton Kivovics** [3], **Tamas Huszar** [1], **Zsolt Nemeth** [1] **and Gyorgy Szabo** [1]

1    Department of Oro-Maxillofacial Surgery and Stomatology, Semmelweis University, 1085 Budapest, Hungary
2    Department of Oral Diagnostics, Semmelweis University, 1088 Budapest, Hungary
3    Department of Community Dentistry, Semmelweis University, 1088 Budapest, Hungary
*    Correspondence: pinter.gabor@phd.semmelweis.hu

**Abstract:** Tantalum and titanium are two of the most popular materials used in dental implants today. These materials are highly biocompatible, durable, and long-lasting, making them ideal for use in dental and maxillofacial implants. The ceramic oxide layer that covers the surface of titanium and tantalum implants ($TiO_2$, $TaO_2$) is formed through an electrochemical growth from the inside of the metal and subsequently altered through heat treatment. The aim of this retrospective study was to evaluate the long-term survival of the oxide ceramic-coated titanium dental implants. The secondary purpose was to evaluate the production process and the cost of the coated tantalum and titanium implants, and to complete these retrospective investigations with the results of our previous work concerning the titanium oxide coating. The structural, physical, and chemical properties as well as the corrosion resistance of the $Ti/TiO_2$ surface were investigated; XPS, SIMS, and XRD techniques were used. Patients who received tantalum oxide-coated ($Ta/TaO_2$) dental implants, titanium oxide-coated ($Ti/TiO_2$) dental implants, or titanium oxide-coated ($Ti/TiO_2$) osteosynthesis plates for rehabilitation at the Department of Oro-Maxillofacial Surgery and Stomatology, Semmelweis University between 1995 and 2005 were included in this retrospective study. Data collection was performed between June 2021 and December 2021. The cost of the tantalum implant was 25 times that of the titanium implant. Only 21 implants were inserted in 10 patients. The survival rate (min. 16 years) was 95%. Twelve patients with a total of sixty-four $TiO_2$-coated implants were observed. The mean implant survival rate was 95%. Our conclusion was that, regardless of the shape of the implant, the $Ti/TiO_2$ coating proved its excellent durability over the years. The "tantalum issue" is increasingly relevant nowadays, since instead of implants made of pure tantalum metal, implants with a porous tantalum surface have come to the fore.

**Keywords:** $TiO_2$; coated titanium implant; tantalum implants; survival; dental implant; maxillofacial implant

## 1. Introduction

Titanium and tantalum are commonly used materials for implants due to their biocompatibility and high strength-to-weight ratio. However, these materials have poor wettability and low osseointegration properties, which can lead to implant failure. To improve the surface properties of titanium and tantalum implants, anodic and thermal oxidation surface modification techniques have been developed.

Tantalum is a rare, reactive metal that has unique properties that make it well suited for dental implants. It is highly biocompatible, meaning that it does not cause any adverse reactions when it comes into contact with living tissues. This makes it ideal for use in dental implants, as it reduces the risk of inflammation and infection.

Titanium is another popular material used in dental implants. Like tantalum, it is biocompatible, durable, and long-lasting. It is also lightweight, making it easier for dentists to work with when installing implants.

From the perspective of improved osseointegration, tantalum has been shown to promote osseointegration, the process by which the implant fuses with the surrounding bone, more effectively than titanium. In relation to better radiopacity, tantalum is more radiopaque, meaning it is more visible on X-rays, which makes it easier for dentists to monitor the placement and progress of the implant. Tantalum is highly corrosion-resistant, making it less likely to degrade over time and increasing the lifespan of the implant. Related to a reduced risk of infection, tantalum is also less prone to infection compared to titanium, which makes it a safer option for dental implant patients.

It is important to note that while tantalum does have several advantages over titanium, titanium remains a highly popular and effective material for dental implants, and the best material for a particular patient will depend on their specific needs and medical history.

Anodic and thermal oxidation are two surface modification techniques that are used to improve the properties of dental implants. These techniques are used to create a ceramic oxide layer on the surface of the implant, which can improve its biocompatibility, stability, and durability.

Anodic oxidation is a surface modification process that involves the application of an electrical current to a titanium or tantalum implant, resulting in the formation of an oxide layer on the surface. This oxide layer improves the wettability of the implant and promotes osseointegration by providing a surface that is more favorable for cell attachment.

Thermal oxidation is another surface modification technique that involves exposing the titanium or tantalum implant to high temperatures in the presence of oxygen, resulting in the formation of an oxide layer on the surface. This oxide layer also improves the wettability and osseointegration of the implant.

The ceramic oxide layer that is produced through anodic or thermal oxidation can provide several benefits for dental implants. The layer can improve the biocompatibility of the implant by reducing the release of metal ions into the surrounding tissues. This can help to reduce the risk of implant failure and minimize the potential for adverse reactions.

In addition, the ceramic oxide layer can improve the stability and durability of the implant by reducing the risk of corrosion and wear. This can help to extend the lifespan of the implant and reduce the risk of complications.

The ceramic oxide layer can also improve the mechanical properties of the implant by increasing its strength and resistance to wear. This can help to reduce the risk of implant failure and improve its long-term performance.

Long-term studies have shown that both anodic and thermal oxidation surface modification techniques result in improved osseointegration and reduced implant failure rates compared to unmodified titanium and tantalum implants. In particular, thermal oxidation has been shown to result in a thicker and more stable oxide layer compared to anodic oxidation, leading to improved long-term performance.

The surface quality of dental and maxillofacial region implants plays a crucial role in determining the connection between the implant and the surrounding tissue [1]. The surface layer or interface must possess certain characteristics, such as:

- it must not alter the underlying bulk properties;
- it must be capable of withstanding chemical, electrical, mechanical, and thermal forces; and
- its properties must remain stable over time.

The spontaneous titanium oxide layer (TiO) is not a true barrier layer [2,3]. After the implantation, titanium ions may be detected in virtually all parts of the organism. The situation would be quite different if a crystalizing, insulating layer was formed on the surface of the titanium. Electrically insulating the oxide layer with a crystalline structure can be produced on metals such as aluminium, titanium, tantalum, and niobium.

Anodizing titanium dental implant surfaces improves the chemical composition and structure of the $TiO_2$ film and enhances the osteogenic response of the osteoblast cells [4,5]. Anodization of implant and abutment surfaces results in a more hydrophilic surface than the turned ones, creating ideal surface characteristics for soft tissue integration [6].

However, there is a lack of evidence for the clinical use of anodized implant surfaces; there are few clinical reports on the long-term performance of anodized implant surfaces [7–13].

Hungarian authors [14,15] elaborated a procedure in which the tantalum plates utilized in various condensers were coated with an oxide layer by means of anodic and thermal treatment, which made these plates passive. In 1992, the same authors developed a similar procedure whereby the surface of titanium implants could be made passive. The procedure was patented [16] and applied successfully in practice [17,18]. The tantalum oxide ($TaO_2$) and titanium oxide ($TiO_2$) layer that was produced had exceptional insulation properties, biocompatibility, and thus was suitable for use in dental and maxillofacial implants.

The ceramic oxide layer on the surface of titanium or tantalum implants is formed electrochemically from the bulk and then modified through heat treatment. The thickness of this layer is influenced by the surface cleanliness of the material, the temperature of the electrolyte, and the voltage applied during the process [14].

The aim of this retrospective study was to evaluate the long-term survival of the oxide ceramic-coated titanium dental implants. The secondary purpose was to evaluate the production process and the cost of the coated tantalum and titanium implants as well as to compare these retrospective investigations with the results of our previous work concerning the titanium oxide coating on dental implants and titanium osteosynthesis plates.

## 2. Materials and Methods

### 2.1. Study Design

The study was conducted in accordance with the Helsinki Declaration. The study protocol was reviewed and approved by the Semmelweis University's Regional Research Ethics Committee (SE RKEB, 212/2021). Patients who received tantalum oxide-coated ($Ta/TaO_2$) dental implants, titanium oxide-coated ($Ti/TiO_2$) dental implants, or titanium oxide-coated ($Ti/TiO_2$) osteosynthesis plates for rehabilitation at the Department of Oro-Maxillofacial Surgery and Stomatology, Semmelweis University between 1995 and 2005 were included in this retrospective study. Data collection was performed between June 2021 and December 2021.

The following clinical and radiological parameters were assessed:

- Presence of absence of suppuration around the implant;
- Presence or absence of plaque;
- Presence or absence of bleeding;
- Presence or absence of periimplantitis.

### 2.2. Production and Coating of the Tantalum and Titanium Implants

The tantalum and titanium implants and their coating were produced by PROTETIM Medical Instrument Ltd., Hódmezővásárhely, Hungary.

The form, length, and diameter of the tantalum and titanium implants were the same: cylindrical form, length between 10–12 mm, and diameter between 3–4 mm.

After milling, implants underwent the following surface treatment:

1. Mechanical and chemical cleaning;
2. Anodic oxidation;
3. Washing and drying;
4. Thermal treatment;
5. Repetition of steps 2–4.

## 3. Results

### 3.1. Patients with Tantalum Oxide-Coated (Ta/TaO$_2$) Dental Implants

Altogether, 21 tantalum dental implants were inserted in 10 patients. The minimum follow-up period was 16 years. Five patients and nine implants could be assessed. The

survival rate was 95%. Note: this number of tantalum implants is not enough for the correct appraisal; we just want to mention the technology in terms of curiosity.

### 3.2. Patients with Titanium Oxide-Coated (Ti/TiO$_2$) Dental Implant

Twelve patients (eight female and four male) who had a total of sixty-four TiO$_2$-coated implants were available for follow up (Table 1). The current age of the patients ranged from 45 to 72 years, with a mean of 67 years. Three of the originally placed sixty-four titanium coated implants were lost, so only data from sixty-one implants can be reported. This means a 95% survival rate (Table 1). No purulent or gingival bleeding was observed from the examined implants. The distance from the implant shoulder to the first bone was 0–4 mm, and the mean distance was 1.2 mm. Inner screw fracture occurred in two cases; the fractured screws were successfully removed, and the prosthetic parts of the implants were renewed.

**Table 1.** Ti/TiO$_2$ implant surviving.

| | Patient | Date of Surgery | Years of Survival | No. of Implants | Complication Failed, Progressive Bone Degradation | Successful Survival |
|---|---|---|---|---|---|---|
| 1 | Patient 1 | 1995 | 26 | 1 | - | 1 |
| 2 | Patient 2 | 1996 | 25 | 6 | - | 6 |
| 3 | Patient 3 | 1995–1997 | 26–24 | 10 | - | 10 |
| 4 | Patient 4 | 1997 | 24 | 3 | Inner screw fracture, solved | 3 |
| 5 | Patient 5 | 2000 | 21 | 10 | 1 failed | 9 |
| 6 | Patient 6 | 2000 | 21 | 2 | - | 2 |
| 7 | Patient 7 | 2001 | 20 | 6 | 1 failed | 5 |
| 8 | Patient 8 | 2001 | 20 | 8 | - | 8 |
| 9 | Patient 9 | 2002 | 19 | 7 | - | 7 |
| 10 | Patient 10 | 2002 | 19 | 2 | - | 2 |
| 11 | Patient 11 | 2005 | 16 | 3 | 1 failed | 2 |
| 12 | Patient 12 | 2005 | 16 | 6 | - | 6 |
| | 12 | 1995–2005 | 16–26 | 64 | 3 failed | 61 |

Survival rate: 95%.

### 3.3. Patients Treated Using Titanium Oxide-Coated (Ti/TiO$_2$) Osteosynthesis Plates

In our previous study [19,20] during 5-year observation period, 108 of 1396 osteosynthesis plates coated with anodic titanium oxide had to be removed for various reasons. Concerning the changes caused in the modified surface by the aggressive action of the organisms, XPS, SIMS, and AES (Auger electron spectroscopy) studies were performed after the removal of the TiO$_2$-coated osteosynthesis plates. The surface provides good protection for the underlying Ti metal [21].

### 3.4. Evaluation of the Production Process and the Cost of the Coated Tantalum and Titanium Implants

After evaluating the production process, it became apparent that the cost of the tantalum implant was 25 times that of the titanium implant due to the difficulty of processing, its much higher density, and its higher base price (Figure 1).

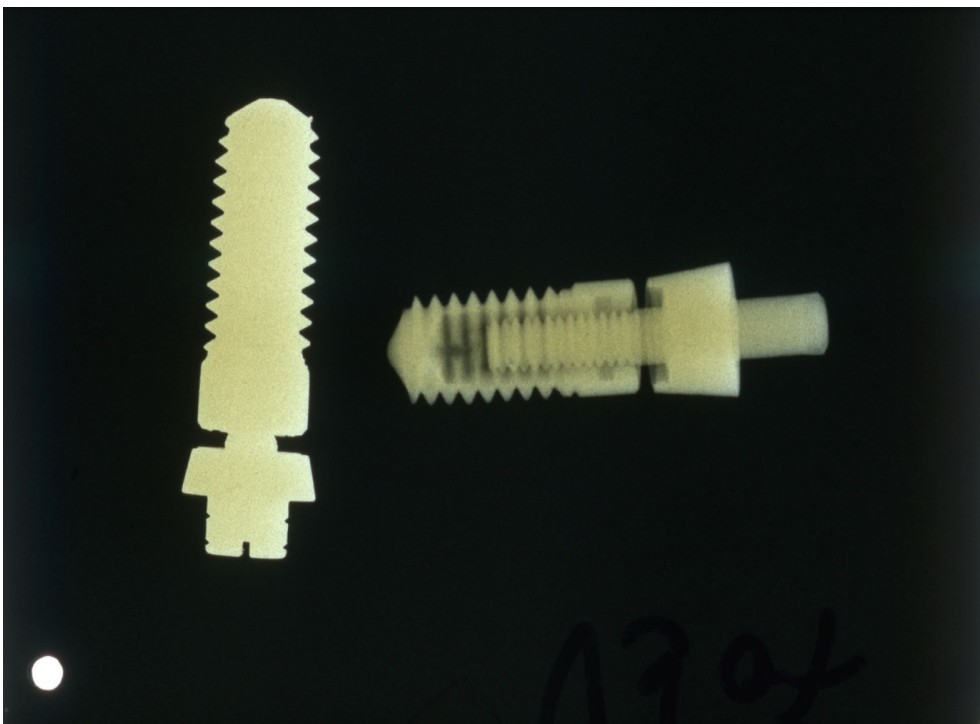

**Figure 1.** X-ray of tantalum (**left**) and titanium (**right**) implants; the density of the tantalum implant is much higher.

## 4. Discussion

In the present study, titanium oxide-coated (Ti/TiO$_2$) osteosynthesis plates and implants showed excellent long term survival rates. Tantalum oxide-coated (Ta/TaO$_2$) dental implants could not be correctly evaluated because a considerable number of the patients were lost for follow-up. The cost of the tantalum implant was 25 times that of the titanium implants. Therefore, only a few implants were placed.

Anodic and thermal oxidation of titanium has been extensively applied to produce implant materials to improve the physical and chemical properties of the bulk metals. To investigate the effect of the surface modification, we previously conducted various examinations not only on dental implants, but on TiO$_2$-coated titanium osteosynthesis plates too.

The structural, physical, and chemical properties of the Ti/TiO$_2$ surface were investigated. The most essential mechanical characteristics of coated and uncoated titanium implants were compared, and these materials were rated according to ISO specifications. The following properties were measured: breaking strength, standard yield points, contraction strength, specific breaking stream, and modules of elasticity. The investigation revealed that coating with the oxide ceramic and thermal treatment of the layer very slightly modify the mechanical properties. The initial ISO categorization of the fundamental material did not need to be changed in these circumstances [19].

X-ray Photoelectron Spectroscopy (XPS) established the chemical composition of the surface; the Ti was present in form of TiO$_2$ [14]. Secondary Ion Mass Spectroscopy (SIMS) provides the possibility for determination of the depth homogeneity of the oxide and for estimating the oxide thickness. The thickness of the oxide was calculated to be 200 nm [20]. Composition of the surface layer was identified via the X-ray diffraction (XRD) technique. It was concluded that the oxide layer is a polycrystalline material made of the anatase polymorph. The corrosion resistance of Ti implants depends mainly on the stability of the oxide layer covering the bulk metal. The time dependence of the open circuit potentials indicated that the TiO$_2$ layer has a high corrosion resistance [22].

During the 5-year observation period, 108 of 1396 plates coated with anodic titanium oxide had to be removed for various reasons. In none of these cases were metalloids observed, which is otherwise reported relatively frequently in the vicinity of traditional titanium-fixing elements, nor was any tissue damage observed surrounding the surface of the plates [21].

The previously listed results were complemented by the current long-term retrospective investigation. The minimum 16 years surviving interval proved the success of such implants coated with $TiO_2$ ceramic. The more-than-90% success rate of osteointegration renders these implants as viable competitors to other "modern" implants with different surface coating systems.

The essence of our work is the titanium oxide ceramic layer. This type of coating, regardless of the shape of the implant (dental implant or osteosynthesis plate), proved its excellent durability over the years. For the sake of comparison, it is worthwhile to mention the article by Jazdzewska and Bartmanski [23] published in 2021 about the nanotubular oxide layer coating of dental screw implants. Oxidation of the Ti-13NG-13Zr titanium alloy was performed. Corrosion tests were performed on the coating, and it was demonstrated that there was a higher corrosion current in general, but less noble corrosion potential due to incomplete surface coverage and pitting.

Regarding the tantalic—$TaO_2$—coating, based on preliminary investigations, it has the same positive characteristics as the Ti/$TiO_2$ ceramic coating. Manufacturing costs of tantalum implants are higher than that of titanium implants, which hinders its widespread use as a dental or maxillo-facial implant material. However, based on the results of the present study, it can be viewed as an alternative to titanium in the manufacturing of dental implants.

There are some (almost) similar techniques for the ceramic coating of dental implants. Novsek et al. [24] reported a method involving a thin layer of anatase on a Ti6Al4V alloy. The coating was composed of pinacoidal anatase grains and strongly attached to the alloy. This layer improves osseointegration, but the authors do not report any long-term investigation. Parnia et al. [25] summarized the application of nanoparticles as Ti-based implant coating materials in order to improve the implant success rate. The authors consider that the $TiO_2$ nanoparticles increase the density of osteoblast cells on the implant, thus enhancing osseointegration and the anticandidal effect. In the review by Dong et al. [26], they recapitulated the existing surface modification technologies of mainstream dental implants and elucidated the correction between implant surface coating and the osseointegration and anti-bacterial ability. The authors concluded that further studies should also examine whether the "mainstream" implant surface treatment and coatings could achieve reliable therapeutic effects, especially osseointegration.

There are various ceramic coatings available for dental implants. Some of the other available materials include bioglasses, other calcium phosphates such as fluorapatite and tricalcium phosphate, and inert ceramics such as alumina. Alternative methods for coating include $TiO_2$ after thermal and electro-galvanic treatment [27]. $TiO_2$ nanotubes have been actively studied as coating materials for implants, and promising results have been reported recently about improving osteogenic activity around implants; indeed, they could be the next step, but they lack long-term investigation [28].

Few retrospective studies report long-term outcomes using dental implants with anodized implant surfaces for prosthodontic rehabilitation. According to their results, dental implants with anodized implant surfaces are a safe and reliable option for implant-supported prostheses with high survival rates [7–13].

Porous Ta implants are already available for clinical use with satisfying results, just like with Ti implants.

Fialho et al. [29] investigated porous tantalum oxide ($Ta_2O_3$) with osteoconductive elements and antibacterial core–shell nanoparticles. The in vitro study of Ta-based implants showed superior biocompatibility by promoting initial cell adhesion and proliferation

as well as antibacterial effects, which are great indicators for osseointegration and the prevention of initial bacterial colonization, respectively.

Piglionico et al. [30] in animal studies investigated the efficiency of porous tantalum dental implants for osseointegration compared with the classical titanium implants. Cell populations on porous tantalum surfaces proliferated more and faster, leading to more calcium deposit production than cells on roughened and smooth titanium surfaces, which revealed a potentially enhanced capacity for osseointegration.

Bencharit et al. [31] summarized the contemporary applications of porous tantalum trabecular metal in implant dentistry. A traditional titanium implant design and equipment were integrated with PTTM technology in the development of PTTM-enhanced titanium dental implants. This potentially allowed for genuine three-dimensional osseointegration augmentation, which might be a significant breakthrough compared to the present concentration on titanium implant surface technology. Based on assumptions from orthopedic clinical trials, the mentioned type of Ta implant may be appropriate in cases of poor healing, immediate/early implant loading, and absent osseous structure necessitating simultaneous implant insertion and bone grafting. The oral cavity's peculiarity, particularly the host–oral microbial relationship, might be a source of worry.

Kim et al. [32] studied whether a dental implant system with a core covered by three-dimensionally porous tantalum material would be more stable than a typical threaded titanium alloy implant system as well as whether bone would grow into the porous region. During 2 to 12 weeks of submerged healing, the porous experimental implants demonstrated a mix of progressive osseointegration (titanium surfaces) and bone ingrowth and maturation inside their porous tantalum portions. Cortical and apical implant threads, in conjunction with the porous tantalum portion, were able to support the experimental implant to the same extent as the completely threaded control implant.

In the review and clinical report of Bencharit et al. [33], it was proposed that extraction and immediate implant surgery may be accomplished effectively even when buccal alveolar bone is lacking. A combination of a demineralized bone matrix, a PTTM-enhanced titanium implant, a custom healing abutment, and an interim partial removable dental prosthesis may have provided an optimal environment for buccal alveolar bone regeneration and osseoincorporation while also preserving the buccal blood supply, enhancing neovascularization, and controlling occlusal loading. Through osseoincorporation, the PTTM provides an optimal environment for initial clot formation, neovascularization, and eventually a scaffold for three-dimensional bone regeneration.

The latter is contradicted to some extent by the animal experiment studies by Fraser et al. [34]. When placed in the rabbit tibia and allowed to heal for at least 4 weeks, the stiffness of the bone-implant interface was similar for threaded implants with or without porous tantalum.

Ayskin Akharzadek et al. [35] evaluated the stress distribution of porous tantalum implants and solid titanium implant-assisted overdenture in the mandible. Applying porous tantalum implants instead of solid titanium implants reduced strain values around both the cortical and trabecular bone.

As a sustainable alternative to replace Ti use in dental implants, tantalum has been pointed out as an alternative because it exhibits bioactivity with good biological responses. Moreover, porous tantalum ($Ta_2O_3$) shows better osteogenic outcomes compared with porous titanium. In addition, ($Ta_2O_3$) shows good antibacterial performance [29].

In the study of Wang H et al. [36], selective laser melting was used to design and fabricate mechanically compatible porous Ta and Ti with the same characteristics. Their biocompatibility, osteoinductivity, and osseointegration were compared. In vitro and in vivo, Ta scaffolds possessed the same mechanical capabilities as trabecular bone and the same inclinations for proliferation, viability, and osteoduction as porous Ti, showing strong mechanical compatibility and biocompatibility. According to the findings, porous Ta is a potential material for bone regeneration.

According to Woo Jin Kim et al. [37] in the case of dental implants, the United States continues to demonstrate titanium-oriented technical advancement, but China is indepen-

dently pursuing tantalum- or zirconium-based material technology. A Chinese patent titled "Method for Preparing Medical Porous Metal Material (CN102796891A)" describes how to make a porous tantalum structure out of tantalum, polyvinyl alcohol, and sodium bicarbonate. The technical trend of employing tantalum as a core material for implants differs greatly from the way tantalum is utilized in other nations, namely for surface treatment of titanium structures [38].

## 5. Conclusions

The results from this long-term/more-than-16-years retrospective study show that anodized surface implants are a safe reliable option for other "modern" implants.

When the manufacturing process was examined, it was discovered that the cost of the pure tantalum implant was 25 times that of the titanium implant. Regardless of the shape of the implant, the $Ti/TiO_2$ coating proved its excellent durability over the years.

Anodic and thermal oxidation are two surface modification techniques that can provide improved properties for tantalum and titanium dental implants. These techniques are used to create a ceramic oxide layer on the surface of the implant, which can improve its biocompatibility, stability, and durability. The choice between anodic and thermal oxidation will depend on the specific requirements of the application, including the desired properties, the budget, and the imaging requirements.

One area in which tantalum and titanium may be improved is in their ability to promote osseointegration, or the process by which the implant fuses with the surrounding bone. New coatings and surface treatments may be developed that increase the speed and efficiency of osseointegration, making implants even more secure and stable.

Another area of focus for the future use of tantalum and titanium in dental implants is their ability to resist wear and tear. As technology advances, new coatings may be developed that help to protect the implant from wear and tear, making it last even longer.

Tantalum and titanium are two of the most popular materials used in dental implants today, and they are likely to remain popular in the future. As technology advances, these materials may become even more advanced and sophisticated, leading to new and improved dental implants that offer even greater benefits to patients. The relevance of the tantalum issue has increased in recent times due to the emergence of implants featuring a porous tantalum surface, rather than those made of pure tantalum metal.

**Author Contributions:** Conceptualization, G.T.P. and G.S.; methodology, G.S.; investigation, G.S.; writing—original draft preparation, G.T.P., B.T. and M.K.; writing—review and editing, G.T.P.; supervision, T.H., Z.N. and G.S.; All authors have read and agreed to the published version of the manuscript.

**Funding:** This research received no external funding.

**Institutional Review Board Statement:** Not applicable.

**Informed Consent Statement:** Not applicable.

**Data Availability Statement:** The data presented in this study are available on request from the corresponding author. The data are not publicly available due to ethical reason.

**Conflicts of Interest:** The authors declare no conflict of interest.

## Abbreviations

| | |
|------|------------------------------------------|
| AES | Auger electron spectroscopy |
| XPS | X-ray Photoelectron Spectroscopy |
| SIMS | Secondary Ion Mass Spectroscopy |
| PTTM | Production of Porous Tantalum Trabecular Metal |

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
