# Peer review of "Long-Term Results of Anodic and Thermal Oxidation Surface Modification on Titanium and Tantalum Implants"

_coatings, doi:10.3390/coatings13040760_

Round 1
Reviewer 1 Report (New Reviewer)
The authors presented an excellent clinical trial on the subject, with appropriate discussion, conclusions based on scientific evidence and extremely relevant to the field of knowledge. There are no further suggestions, other than congratulating the authors on the excellent work.
Author Response
Dear Reviewer,
I am writing to express my sincere gratitude for your time and effort in reviewing our manuscript for Coatings. Your thoughtful comments and constructive feedback were invaluable in improving the quality and clarity of our work.
Sincerely
Reviewer 2 Report (New Reviewer)
This is a unique paper examining long term implantation results of a metallic treatment method that the authors developed some years ago. The results are long term, although not many implants were placed. I support publication after revision.
Methods:
More references to the older papers on material/surface synthesis are needed.
Results:
Line 101 to 102 – I don’t understand what the authors mean.
Discussion:
Is there more information available on the etiology of failure?
Are there other demographic/patient information available that would help elucidate the etiology?
Some of the short paragraphs in the discussion could be combined for better flow.
Author Response
Dear Reviewer,
Thank you for taking the time to review our manuscript and for providing us with valuable feedback. We are writing to inform you that we have made the requested revisions and have resubmitted the manuscript accordingly.
We carefully considered your comments and suggestions and have incorporated them into the revised version of the manuscript. We are confident that the changes we have made have significantly improved the clarity and quality of our work.
Once again, we thank you for your constructive feedback and for giving us the opportunity to improve our manuscript. We appreciate your efforts in helping us to make our work better.
Please let us know if you require any further information or have any additional concerns.
Sincerely,
Reviewer 3 Report (New Reviewer)
In the abstract Authors mention many aspects of the work, none of which was properly presented in the given article. First of all, there are no results collected between June 2021 and December 2021.
Both Introduction and Materials and Methods paragraphs are written in very general terms. Chapters 2.1. and 2.2. do not describe how the process was actually conducted.
I do not understand what the Authors wanted to present in this work. Authors describe, also vaguely, that certain experiments were carried out and certain values were measured, but this is not presented in the paper.
The Authors also try to refer to their results from more than 15 years ago, without comparing them in any way to what they obtained in 2021, as they describe in the abstract.
Authors do not compare their results with those of other research groups. Conclusions are very general. The Authors included three sentences that are not confirmed in the materials presented in the article (the cost of producing the implant, why it is a safe and reliable option). To which modern implants do the authors compare their results?
The Authors present, in the form of a table, information for only 12 patients, for whom the survival of 61 implants was verified. Results for other implants (including those described in paragraph 3.1) are not presented. Section 3.3 does not present the actual results in this work. The results are probably in the Authors' previous publications. Apart from a short paragraph, there is no information, or results about plate-shaped implants.
I do not understand where the Authors took the production costs in which they write in paragraph 3.4. There is no specific information about this anywhere in the publication. However, the Authors mention that they conducted an evaluation of the production process.
What is the "tantalum issue"?
The work also seems to be chaotic and messy considering, among other things, lines 120-124. I guess that they are unnecessary and have not been removed by the Authors at the stage of preparing the article.
In paragraph 2.2. the shape of the implant is not described, only a cylindrical shape is mentioned. There is no information about the exact shape of the prepared implants, it can only be deduced from Fig. 1.
The Authors present a lack of consistency in the description, i.a. in paragraph 3.2. there is information "TiO2 coated", then "titanium coated" but as I understand it refers to the same implants. Titanium oxide is not the same as titanium.
The work is written in a clear, linguistically correct way. However, the Authors did not avoid a few grammatical errors, inappropriate singular/plural forms, etc., for example in line 44-46, 47-48.
There are mistakes in spelling of et al. abbreviation, and spelling of chemical compounds (i.a. in line 48, 59, 76).
Author Response
Dear Reviewer,
Thank you for taking the time to review our manuscript and for providing us with valuable feedback. We are writing to inform you that we have made the requested revisions and have resubmitted the manuscript accordingly.
We carefully considered your comments and suggestions and have incorporated them into the revised version of the manuscript. We are confident that the changes we have made have significantly improved the clarity and quality of our work.
Once again, we thank you for your constructive feedback and for giving us the opportunity to improve our manuscript. We appreciate your efforts in helping us to make our work better.
Please let us know if you require any further information or have any additional concerns.
Sincerely,
Reviewer 4 Report (New Reviewer)
Dear Authors,
you made a great work! However, some improvements are suggested.

Author Response
Dear Reviewer,
Thank you for taking the time to review our manuscript and for providing us with valuable feedback. We are writing to inform you that we have made the requested revisions and have resubmitted the manuscript accordingly.
We carefully considered your comments and suggestions and have incorporated them into the revised version of the manuscript. We are confident that the changes we have made have significantly improved the clarity and quality of our work.
Once again, we thank you for your constructive feedback and for giving us the opportunity to improve our manuscript. We appreciate your efforts in helping us to make our work better.
Please let us know if you require any further information or have any additional concerns.
Sincerely,
Round 2
Reviewer 2 Report (New Reviewer)
The authors made all changes as I requested. I appreciate. However, in the future, it would be easier for the reviewers to include the point by point document and not just a generic cover letter.
Reviewer 3 Report (New Reviewer)
I endorse the manuscript for publication. It was corrected in accordance with the review.
Reviewer 4 Report (New Reviewer)
Dear Authors,
I believe this manuscript is suitable for publication.
This manuscript is a resubmission of an earlier submission. The following is a list of the peer review reports and author responses from that submission.
Round 1
Reviewer 1 Report
Article "Long-term results of anodic and thermal oxidation surface
modification on titanium and tantalum implants" needs significant revision, because:
1. The main paragraphs "Introduction", "Materials and Methods", "Results and Discussion" are designed in a non-standard way. The introduction does not state the problem, there are no studies on the subject of the article, and the purpose of the study is not formulated. In the discussion of the results, reference is made to the high corrosion properties of the obtained coatings to articles by other authors.
2 In addition to Table 1, what other original results are available that would confirm the results obtained?
3 There are typos "Artivle", "TO2" and others in the text
4 Abstract needs revision
After a significant revision of the article, it is possible to return to the review of the updated article.
Reviewer 2 Report
The manuscript "Long-term results of anodic and thermal oxidation surface modification on titanium and tantalum implants" is not presented in a well-structured manner. All sections of this manuscript demand major revision. It is very hard to understand what, why and how experiments were performed. Cited literature is mostly very old besides the fact that 9 out of 15 references are from the authors' group.
Reviewer 3 Report
The author has reported paper on Long-term results of anodic and thermal oxidation surface 2 modification on titanium and tantalum implants. but there are many issues with article regarding introduction result and presentation. at present form not accepted.